# DPPNET: Approximating Determinantal Point Processes with Deep Networks

**Zelda Mariet** [*]
Massachusetts Institute of Technology
Cambridge, Massachusetts 02139, USA
zelda@csail.mit.edu

**Yaniv Ovadia & Jasper Snoek**
Google Brain
Cambridge, Massachusetts 02139, USA
{yovadia, jsnoek}@google.com

## Abstract

Determinantal point processes (DPPs) provide an elegant and versatile way to sample sets of items that balance the quality with the diversity of selected items. For this reason, they have gained prominence in many machine learning applications that rely on subset selection. However, sampling from a DPP over a ground set of size $N$ is a costly operation, requiring in general an $\mathcal{O}(N^3)$ preprocessing cost and an $\mathcal{O}(Nk^3)$ sampling cost for subsets of size $k$. We approach this problem by introducing DPPNETs: generative deep models that produce DPP-like samples for arbitrary ground sets. We develop an inhibitive attention mechanism based on transformer networks that captures a notion of dissimilarity between feature vectors. We show theoretically that such an approximation is sensible as it maintains the guarantees of inhibition or dissimilarity that makes DPPs so powerful and unique. Empirically, we show across multiple datasets that DPPNET is orders of magnitude faster than competing approaches for DPP sampling, while generating high-likelihood samples and performing as well as DPPs on downstream tasks.

## 1 Introduction

Selecting a representative sample of data from a large pool of available candidates is an essential step of a large class of machine learning problems: noteworthy examples include automatic summarization, matrix approximation, and minibatch selection. Such problems require sampling schemes that calibrate the tradeoff between the point-wise *quality* – *e.g.* the relevance of a sentence to a summary – of selected elements and the set-wise *diversity*[2] of the sampled set.

Submodular set functions and their log-submodular counterparts (functions $f$ such that $\log f$ is submodular) have arisen as a theoretically grounded model for such diversity modeling problems, with applications to settings such as sensor placement [27], summarization [33], and optimal experimental design [44]. Submodular functions over a ground set $[N] := \{1, \ldots, N\}$ are functions $f : 2^{[N]} \to \mathbb{R}$ that satisfy the inequality

$$f(S) + f(T) \geq f(S \cap T) + f(S \cup T) \quad \text{for } S, T \subseteq [N].$$

Among log-submodular measures, determinantal point processes (DPPs) have proven to be of particular interest to the machine learning community, due to their ability to elegantly model the tradeoff between quality and diversity. Given a ground set of size $N$, DPPs allow for $\mathcal{O}(N^3)$ sampling over all $2^N$ possible subsets of elements, assigning to any subset $S$ of elements the probability

$$\mathcal{P}_{\boldsymbol{L}}(S) = \det \boldsymbol{L}_S / \det(\boldsymbol{I} + \boldsymbol{L}), \tag{1}$$

---

[*]Work done while at Google Brain.

[2]Here, we use diversity to mean useful coverage across dissimilar examples in a meaningful feature space, rather than other definitions of diversity that may appear in the ML fairness literature.

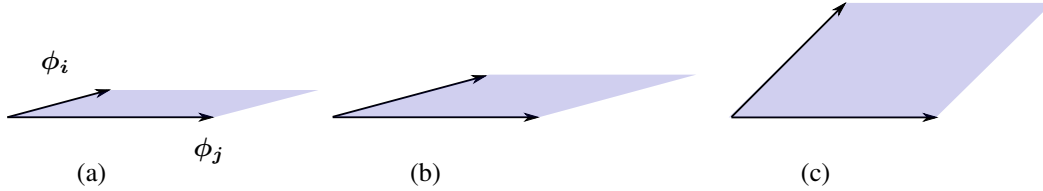

Figure 1: Geometric intuition for DPPs: let $\phi_i, \phi_j$ be two feature vectors of $\mathbf{\Phi}$ such that the kernel verifies $\boldsymbol{L} = \mathbf{\Phi}\mathbf{\Phi}^T$; then $\mathcal{P}_{\boldsymbol{L}}(\{i,j\}) \propto \text{Vol}(\phi_i, \phi_j)^2$. Increasing the norm of a vector (quality) or increasing the angle between the vectors (diversity) increases the spanned volume [28].

where $\boldsymbol{L} \in \mathbb{R}^{N \times N}$ is the DPP kernel and $\boldsymbol{L}_S = [\boldsymbol{L}_{ij}]_{i,j \in S}$ denotes the principal submatrix of $\boldsymbol{L}$ indexed by items in $S$ (we adopt here the $\mathcal{L}$-Ensemble construction [7] of a DPP). Intuitively, DPPs measure the volume spanned by the feature embedding of the items in feature space (Figure 1).

First introduced by Macchi [35] to model the distribution of possible states of fermions obeying the Pauli exclusion principle, the properties of DPPs have since then been studied in depth, *e.g.*, [24, 6]. As DPPs capture repulsive forces between similar elements, they arise in many natural processes, such as the distribution of non-intersecting random walks [25], spectra of random matrix ensembles [41, 17], and zero-crossings of polynomials with Gaussian coefficients [23]. More recently, DPPs have become a prominent tool in machine learning due to their elegance and tractability over small datasets: recent applications include video recommendations [9], minibatch selection [51], kernel approximation [31, 38], and neural network pruning [36]; continuous DPPs have also been connected to active learning [22].

However, $\mathcal{O}(N^3)$ sampling makes DPPs intractable for large datasets. This has led to the development of alternate approaches such as subsampling from $\{1, \ldots, N\}$, structured kernels [15, 37], tree-based samplers [16] and approximate sampling [2, 30, 1]. While faster than the standard approach, these methods require significant pre-processing time or cannot be parallelized, and still scale poorly with the size of the dataset. Furthermore, when dealing with ground sets with variable components, pre-processing costs cannot be amortized, impeding the application of DPPs in practice. Recently, Dereziński et al. [12] showed that for exact sampling, the preprocessing cost can be reduced to $\mathcal{O}(N\text{poly}(k))$, where $k$ is the size of the sampled set.

These setbacks motivate us to investigate more scalable and flexible models to generate high-quality, diverse samples from datasets. We introduce generative deep models to approximate the DPP distribution over a ground set of items with both fixed and variable feature representations. We show that a carefully constructed neural network, DPPNET, can generate DPP-like samples with little overhead, orders of magnitude faster than all competing approaches. We further motivate our approach by proving that neural networks are theoretically able to inherit the log-submodularity properties of their target functions. Finally, we show that DPPNETs can trivially approximate conditional DPP samples and greedy mode finding.

## 2 Related work

Although the greedy maximization of submodular and log-submodular set functions is possible with provable guarantees under a variety of constraints [27], sampling and evaluating submodular functions is not necessarily computationally feasible. Indeed, approximating submodular functions has been studied in discrete optimization and game theory [4, 13]; approximate sampling for log-submodular functions has also been considered by Gotovos et al. [19] via MCMC sampling schemes.

In the general case, sampling exactly from a DPP over a discrete set of $N$ items requires an initial eigendecomposition of the kernel matrix $\boldsymbol{L}$, incurring a $\mathcal{O}(N^3)$ cost. In order to avoid this time-consuming step, several approximate sampling methods have been derived; Affandi et al. [1] approximate the DPP kernel during sampling; more recently, results by Anari et al. [2] followed by Li et al. [30] showed that DPPs are amenable to efficient MCMC-based sampling methods.

Exact methods that significantly speed up sampling by leveraging specific structure in the DPP kernel have also been developed [37, 15, 43, 39]. Of particular interest is the dual sampling method introduced in Kulesza and Taskar [28]: if the DPP kernel can be composed as an inner product over a finite basis, i.e. there exists a feature matrix $\mathbf{\Phi} \in \mathbb{R}^{N \times D}$ such that the DPP kernel is given by $\boldsymbol{L} = \mathbf{\Phi}\mathbf{\Phi}^{\top}$, exact sampling can be done in $\mathcal{O}(ND^2 + NDk^2 + D^2k^3)$.

However, MCMC sampling requires variable amounts of sampling rounds, which is unfavorable for parallelization; dual DPP sampling requires an explicit feature matrix $\mathbf{\Phi}$. Motivated by recent work on modeling set functions with neural networks [50, 10], we propose to generate approximate samples via a generative network; this allows for simple parallelization while simultaneously benefiting from recent improvements in specialized architectures for neural network models (*e.g.* parallelized matrix multiplications). We furthermore show that, extending the abilities of dual DPP sampling, neural networks may take as input variable feature matrices $\mathbf{\Phi}$ and sample from non-linear kernels $\mathbf{L}$.

# 3 Generating DPP samples with deep models

In this section, we describe our approach to generating approximate DPP samples using a generative neural network; by doing so, we avoid the $\mathcal{O}(N^3)$ computational cost of DPP sampling, generating samples orders of magnitude faster than competing approaches.

Our goal is to generate samples from a ground set $\{1, \ldots, N\}$ where each item $i$ is represented by a feature $\phi_i \in \mathbb{R}^d$. Although in select cases we may know the related feature matrix $\mathbf{\Phi}$ a priori, in many situations $\mathbf{\Phi}$ will evolve over time. For example, this is the case when $\mathbf{\Phi}$ represents a pool of products that are available for sale at a given time, or social media posts whose relevance varies based on context. For this reason, $\mathbf{\Phi}$ is considered to be an input to our model. Figure 2 presents the architecture of our model.

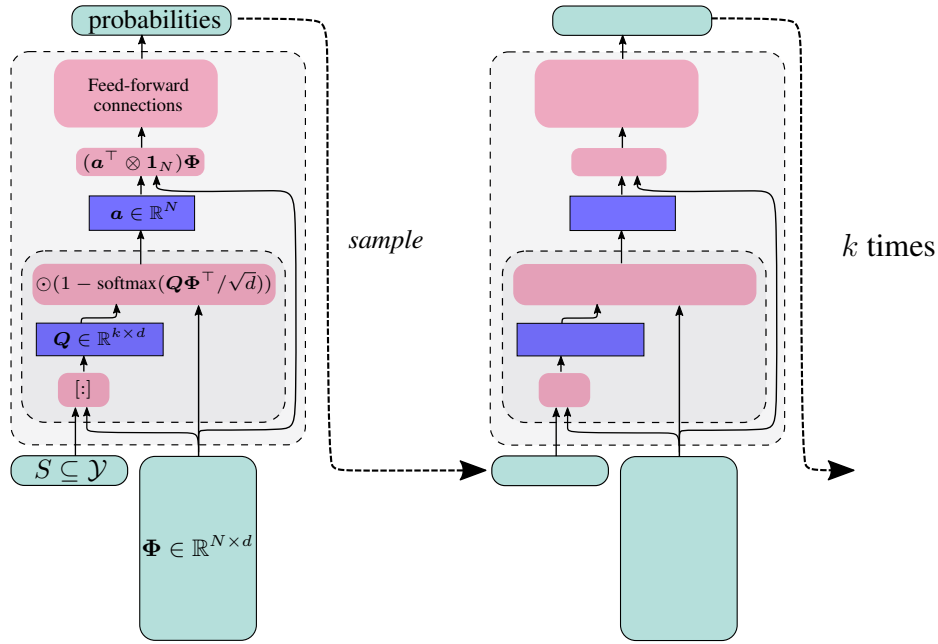

Figure 2: DPPNET takes as input a set $S$ (one-hot encoding) and a representation $\mathbf{\Phi} \in \mathbb{R}^{N \times d}$ of the ground set, and outputs a probability vector of length $N$ representing the respective probabilities of adding any item $i$ to $S$. When initialized with the empty set and repeated $k$ times, this process generates a sample of size $k$. When the feature representation $\mathbf{\Phi}$ does not evolve over time, DPPNET consists only of the block of feed-forward connections and ignores $\mathbf{\Phi}$ as an input.

## 3.1 Motivating the model choice

In addition to elegantly capturing the quality/diversity tradeoffs that arise in subset sampling tasks within machine learning, DPPs also enjoy a variety of properties which make them particularly amenable to modeling via neural networks. Conversely, we may want to preserve some of the properties of the standard DPP sampling algorithm. In this section, we show how such properties of DPPs are either preserved by DPPNET or can be efficiently incorporated into it.

**Simple computation of marginal probabilities.** Given a set $Y$ sampled by a DPP with kernel $\mathbf{L}$ and $S \subseteq Y$, the probability $\Pr(\{i\} \cup S \subseteq Y \mid S \subseteq Y)$ of also sampling $i \notin S$ has the closed form

$$\Pr(\{i\} \cup S \subseteq Y \mid S \subseteq Y) = 1 - [(\mathbf{L} + \mathbf{I}_{[N] \setminus S})^{-1}]_{ii}. \tag{2}$$

Although Eq. 2 requires an expensive matrix inversion to compute, such costs can be offset during off-line training. These probabilities act as the vector of probabilities that DPPNET seeks to output. This signal has the advantage of providing information during every step of training, compared to other downstream possibilities (*e.g.*, the negative log-likelihood of generated sets). In practice, we found the the L1 loss led to the best performance; hence, DPPNET is trained to minimize the L1 distance from its output vector to the corresponding normalized probabilities of adding an item.

**Sequential sampling.** The standard DPP sampling algorithm [28] generates samples sequentially, adding items one after the other until reaching a pre-determined size[3]. We take advantage of this by recording all intermediary subsets generated by the DPP when sampling training data.

In practice, instead of training on $n$ subsets of size $k$, we train on $kn$ subsets of size $0, \dots, k-1$.

The sequential form of exact sampling is also amenable to simple modifications that yield greedy sampling algorithms [8]. For this reason, our architecture also implements sequential sampling (Alg. 1), yielding a straightforward greedy estimation without further overhead.

---

**Algorithm 1** Sampling and greedy mode for DPPNET

---

**Input:** Initial set $S$, target size $k$, feature matrix $\mathbf{\Phi}$
**while** $|S| < k$ **do**
    $v \leftarrow \text{DPPNET}(S, \mathbf{\Phi})$
    **if** sampling **then**
        $i \sim \text{Multinomial}(v/\|v\|)$
    **else if** greedy mode **then**
        $i \leftarrow \text{argmax}\, v$
    $S \leftarrow S \cup \{i\}$
**return** $S$

---

**Closure under conditioning.** DPPs are closed under conditioning: given $A \subseteq \mathcal{Y}$, the conditional distribution over $\mathcal{Y} \setminus A$ given by $\mathcal{P}_L(S = A \cup B \mid A \subseteq S)$ for $B \cap A = \emptyset$ is a DPP with kernel $\mathbf{L}^A = ([(\mathbf{L} + \mathbf{I}_{\bar{A}})^{-1}]_{\bar{A}})^{-1} - \mathbf{I}$ (see [7]). This property make DPPs well-suited to applications requiring diversity in conditioned sets, *e.g.* basket completion tasks.[4]

Standard deep generative models such as (Variational) Auto-Encoders [26] (VAEs) and Generative Adversarial Networks [18] would not enable conditioning operations during sampling, as such operations would have to take place over the model's latent space. With the DPPNET architecture, we can sample a set via Alg. 1, which allows for trivial basket-completion type conditioning operations.

### 3.2 The inhibitive attention mechanism: sampling over variable feature matrices

In simple settings, we wish to draw samples over a ground set with fixed features $\mathbf{\Phi}$. In this case, DPPNET's knowledge of $\mathbf{\Phi}$ can be obtained during training, and so DPPNET is a feed-forward network taking a partially sampled set $S$ as input. However, often the feature representation $\mathbf{\Phi}$ will vary across time or contexts. In such cases, DPPNET also takes as input the feature matrix $\mathbf{\Phi}$.

We confirmed that naively adding $\mathbf{\Phi}$ as input to a stack of feed-forward connections requires deeper networks and larger layers, increasing learning and sampling time. Instead, we draw inspiration from the dot-product attention of [47]. Intuitively, attention is a vector computed by the network that indicates relevant parts of the inputs. For DPPNET, this attention *reweights* $\mathbf{\Phi}$ based on items in $S$.

In [47], the attention mechanism takes three matrices as input, which can be viewed as a (1) the keys $\mathbf{K}$, (2) the values $\mathbf{V}$, and (3) a the query $\mathbf{Q}$. The attention matrix $\mathbf{A} := \text{softmax}(\mathbf{Q}\mathbf{K}^\top/\sqrt{d})$ reweights the values $\mathbf{V}$, with $d$ is the dimension of each query/key and the softmax being computed across each row. The inner product[5] acts as a proxy to the similarity between queries and keys.

For DPPNET, the submatrix of $\mathbf{\Phi}$ given by $\mathbf{\Phi}_{S,:} \in \mathbb{R}^{|S| \times d}$ and corresponding to the items in the input set $S$ acts as the query; the representation $\mathbf{\Phi} \in \mathbb{R}^{N \times d}$ of the ground set is both the keys and the values. In order for the attention mechanism to make sense in the framework of DPP modeling, we make two modifications to the attention in [47]:

- DPPNET should attend to items that are *dissimilar* to those in input subset $S$: for $i \in S$, we compute its pairwise dissimilarity to all items in $\mathcal{Y}$ as the vector $\mathbf{d}_i = 1 - \text{softmax}(\mathbf{\Phi}_{i,:}\mathbf{\Phi}^\top/\sqrt{d})$.

- Instead of returning the $k \times N$ matrix of dissimilarities, we return a vector $\boldsymbol{a} \in \mathbb{R}^N$ in the probability simplex such that $a_j \propto \prod_{i \in S} \boldsymbol{d}_{ij}$. This yields a fixed-size input to the network; this forces the similarity of any item $j$ to a *single* pre-sampled item $i$ to disqualify $j$ from being sampled.

With these modifications, our attention vector $\boldsymbol{a}$ is computed via the *inhibitive attention* mechanism

$$\boldsymbol{a}' = \underset{i \in S}{\odot}\left(1 - \operatorname{softmax}(\boldsymbol{\Phi}_{i,:}\boldsymbol{\Phi}^\top/\sqrt{d})\right), \qquad \boldsymbol{a} = \boldsymbol{a}'/\|\boldsymbol{a}'\|_1, \tag{3}$$

where $\odot$ represents the row-wise multiplication; $\boldsymbol{a}$ can be computed in $\mathcal{O}(kDN)$ time. The attention $\boldsymbol{a}$ is finally multiplied element-wise with each row of $\boldsymbol{\Phi}$; the resulting reweighted feature matrix is the input to the feed-forward component of DPPNET.

**Remark 1.** An efficient ("dual") DPP sampling algorithm for kernels of the form $\boldsymbol{L} = \boldsymbol{\Phi}\boldsymbol{\Phi}^\top$ was introduced in [28]. However, this algorithm requires knowledge of such low-rank decomposition. For non-linear kernels, a low-rank decomposition of $\boldsymbol{L}(\boldsymbol{\Phi})$ must first be obtained, requiring $\mathcal{O}(N^3)$ time. In comparison, the dynamic DPPNET models DPPs with kernels that depend arbitrarly on $\boldsymbol{\Phi}$, including kernels with kernel functions too costly to be computed on-demand.

### 3.3 Sampling over ground sets of varying size

When the ground set size $N$ is expected to vary little over time (*e.g.*, recommender systems where available items are added/removed over time in small numbers), we can modify the architecture of Fig. 2 by slightly overshooting the number of rows $N'$ of the feature matrix $\boldsymbol{\Phi}$ so as to guarantee $N \leq N'$. By setting the additional $N' - N$ rows of $\boldsymbol{\Phi}$ to 0, as well as the $N' - N$ coefficients of the output probability vector, we maintain DPPNET properties and allow variations in ground set size.

When $N$ has high variance, the inhibitive attention mechanism can be modified to accomodate subsampling: a subset $T \subseteq [N]$ of pre-defined size is selected by sampling $|T|$ items independently from the distribution parametrized by the attention vector $\boldsymbol{a}$. The corresponding fixed-size feature matrix is reweighted by the attention, then fed to the learnable feed-forward network. Note that this approach can be combined or replaced by other subsampling schemes for DPPs, *e.g.*, [11].

### 3.4 Preserving log-submodularity

A fundamental property of DPPs is their log-submodularity. Indeed, log-submodularity is one of the few key properties responsible for DPPs's preference for diverse subsets [5].This section presents a surprising result: under certain conditions, the (log) submodularity of a distribution $\mathcal{P}$ can be inherited by a generative model trained to approximate $\mathcal{P}$. In particular, DPPNET may under the right conditions generate samples from a log-submodular distribution. The proof of Theorem 1 can be found in Appendix A.

**Theorem 1.** *Let $f : 2^{\mathcal{Y}} \to \mathbb{R}$ be a strictly submodular function over subsets of a ground set $\mathcal{Y}$, and let $g$ be a function over the same space such that*

$$\|f - g\|_\infty \leq \min_{\substack{S \neq T, \\ S,T \notin \{\emptyset, \mathcal{Y}\}}} \tfrac{1}{4}\left[f(S) + f(T) - f(S \cup T) - f(S \cap T))\right].$$

*Then $g$ is also submodular.*

**Remark 2.** Thm. 1 can also be stated for supermodular functions.

Cor. 1 for DPPNET follows directly from the equivalence of norms in finite dimensional spaces.

**Corollary 1.** *Let $\mathcal{P}_{\boldsymbol{L}}$ be a strictly log-submodular DPP over $\mathcal{Y}$; let DPPNET be a network trained with loss function $\|\boldsymbol{p} - \boldsymbol{q}\|$, where $\|\cdot\|$ is a norm and $\boldsymbol{p} \in \mathbb{R}^{2^N}$ (resp. $\boldsymbol{q}$) is the probability vector assigned by the DPP (resp. the DPPNET) to each subset of $\mathcal{Y}$. Let $\alpha = \max_{\|\boldsymbol{x}\|_\infty = 1} 1/\|\boldsymbol{x}\|$. The distribution modeled by DPPNET is log-submodular if its loss satisfies*

$$\|\boldsymbol{p} - \boldsymbol{q}\| \leq \min_{\substack{S \neq T \\ S,T \notin \{\emptyset, \mathcal{Y}\}}} \tfrac{1}{4\alpha}\left[\mathcal{P}_{\boldsymbol{L}}(S) + \mathcal{P}_{\boldsymbol{L}}(T) - \mathcal{P}_{\boldsymbol{L}}(S \cup T) - \mathcal{P}_{\boldsymbol{L}}(S \cap T))\right].$$

**Remark 3.** Cor. 1 is generalizable to the KL divergence loss $D_{\text{KL}}(\mathcal{P}\|\mathcal{Q})$ via Pinsker's inequality.

Checking numerically whether the conditions for Corollary 1 apply during training is NP-hard: the results of this section are purely theoretical. However, Theorem 1 and Corollary 1 provide an additional justification for the use of probabilities in the objective function of DPPNET, compared to other possible choices for the loss (such as the NLL of generated subsets).

# 4 Experimental results

To evaluate DPPNET, we evaluate its performance (a) as a proxy for a static DPP (with fixed kernel $L$) and (b) a generator of diverse subsets of varying ground sets. Our models are trained with TensorFlow using the Adam optimizer. Hyperparameters are tuned to maximize the normalized log-likelihood of generated subsets. We compare DPPNET to standard DPPs and to the following baselines:

- UNIF: Uniform sampling over the ground set,
- HCP: Matérn hard core point processes. Points are sampled from a Poisson distribution then thinned out to remove points within distance $r < 0.2$ (chosen by cross-validation) from each other,
- $k$-MEDOIDS: The clustering algorithm from [21], which uses datapoints as cluster centers. The distance between points is computed using the same metric used by the DPP.

In sections 4.1 and 4.2 we evaluate the quality of training and ability of DPPNET to emulate DPP samples. For this reason, we evaluate subset quality using the subset's negative log-likelihood (NLL) under the DPP we seek to approximate, as – to the extent of our knowledge – there is no other standard method to benchmark the diversity of a selected subset that depends on specific dataset encodings.

In section 4.3, we evaluate DPPNET sampling as a proxy for DPP samples on a downstream task (kernel reconstruction); there, the evaluation metric evaluates the quality of the reconstructed kernel.

## 4.1 Sampling over the unit square

We begin by analyzing the performance of a DPPNET trained on a DPP with fixed kernel over the unit square. This is motivated by the need for diverse sampling methods on the unit hypercube, *e.g.* quasi-Monte Carlo methods, latin hypercube sampling [40] and low discrepancy sequences.

The ground set consists of the 100 points lying on the $10 \times 10$ grid on the unit square. The DPP is defined by its kernel $L$ such that $L_{ij} = \exp(-\|x_i - x_j\|_2^2/2)$. As the target distribution has a fixed ground set representation (by way of $L$), DPPNET has no inhibitive attention mechanism. We report the performance of the different sampling methods in Figure 3. Visually (Figure 3a) and quantitively (Figure 3b), DPPNET improves significantly over all other baselines. Furthermore, greedily sampling the mode from the DPPNET achieves a better NLL than DPP samples themselves (Table 1).

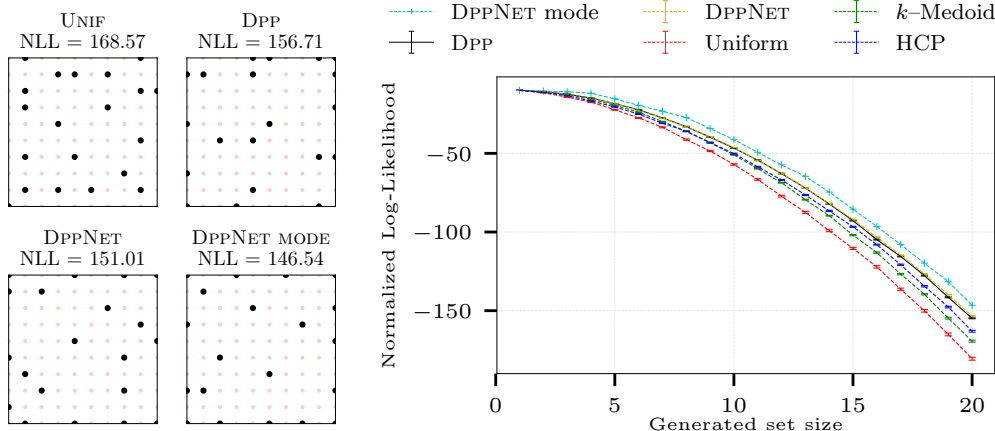

(a) Sampled subsets of size 20 and corresponding NLLs for several baselines.

(b) Normalized log-likelihood of samples drawn from all methods as a function of the sampled set size.

Figure 3: Sampling on the unit square with a DPPNET (1 hidden layer, 841 neurons) trained on a single DPP kernel. Visually, DPPNET gives similar results to the full DPP (left). As evaluated by DPP NLL, the DPPNET's mode achieves superior performance to the full DPP, and DPPNET sampling overlaps with DPP sampling (right).

Table 1: Negative log likelihood (NLL) under $\mathcal{P}_L$ for sets of size $k = 20$ sampled over the unit square. DPPNET achieves comparable performance to the DPP, outperforming the other baselines. DPP GREEDY is deterministic greedy DPP sampling and achieves the lowest NLL; however, DPPNET MODE is able to reach it (Fig. 3a).

| DPP | DPP GREEDY | UNIFORM | HCP | $k$-MEDOIDS | DPPNET |
|---|---|---|---|---|---|
| $154.95 \pm 2.93$ | $147.76$ | $180.53 \pm 9.56$ | $163.40 \pm 5.87$ | $169.37 \pm 6.41$ | $153.44 \pm 2.07$ |

## 4.2 Sampling over variable ground sets

We evaluate the performance of DPPNETs on varying ground set sizes through the MNIST [29], CelebA [34], and MovieLens [20] datasets. For MNIST and CelebA, we generate feature representations of length 32 by training a VAE on the dataset (see App. B for details); for MovieLens, we obtain a feature vector for each movie by applying nonnegative matrix factorization the rating matrix, obtaining features of length 10. We train DPPNET with the embeddings corresponding to randomly subsampled ground sets of size $N = 100$ of the training sets of each dataset; during testing (*i.e.*, in the results below), the trained models are fed feature representations from the corresponding test sets.

The DPPNET is trained based on samples from DPPs with a linear kernel for MovieLens and with an exponentiated quadratic kernel for the image datasets. Bandwidths were set to $\beta = 0.0025$ for MNIST and $\beta = 0.1$ for CelebA, chosen in order to obtain a DPP sample size $\approx 20$: for a DPP with kernel $\boldsymbol{L}$, the expected sample size is given by $\mathbb{E}_{S \sim \mathcal{P}_L}[|S|] = \mathrm{Tr}[\boldsymbol{L}(\boldsymbol{L} + \boldsymbol{I})^{-1}]$.

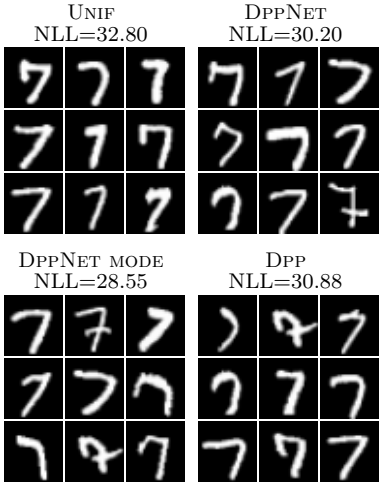

|  |  |
|---|---|
| UNIF NLL=32.80 | DPPNET NLL=30.20 |
| DPPNET MODE NLL=28.55 | DPP NLL=30.88 |

Figure 4: Digits sampled from a DPPNET (3 layers of 365 neurons) trained on MNIST.

For MNIST, Figure 4 shows images selected by the baselines and the DPPNET, chosen among 100 digits with all identical labels; visually, DPPNET and DPP samples provide a wider coverage of writing styles. However, the NLL of samples from DPPNET decay significantly, whereas the DPPNET mode maintains competitive performance with DPP samples. For this reason, all further experiments focus on greedy mode samples drawn from the DPPNET.

Numerical results for MNIST are reported in Table 2. Although DPPNET was trained on feature matrices representing random subsets of the training set, we see that when selecting subsets restricted by label at test time, DPPNET remains competitive, suggesting that DPPNET sampling may be leveraged to focus on sub-areas of datasets identified as areas of interest. Numerical results for CelebA and MovieLens are reported in Table 3.

To analyze the contribution of the attention mechanism, we furthermore performed an ablation test, training a neural network without the attention block; architecture tuning revealed that the model that achieved the best performance required 6 layers of 585 neurons on MNIST: significantly more parameters than with the attention mechanism (3 layers of 365 neurons).

Table 2: NLL (mean $\pm$ standard error) under the true DPP of samples drawn uniformly, according to the mode of the DPPNET, and from the DPP itself. We sample subsets of size 20; for each class of digits we build 25 feature matrices $\boldsymbol{\Phi}$ from encodings of those digits, and for each feature matrix we draw 25 different samples. For the last column, DPPNET was trained on all digits.

| DIGIT | 0 | 1 | 2 | 3 | 4 |
|---|---|---|---|---|---|
| DPP BASELINE | $52.2 \pm 0.1$ | $60.5 \pm 0.1$ | $49.8 \pm 0.0$ | $50.7 \pm 0.1$ | $51.0 \pm 0.1$ |
| UNIF | $54.9 \pm 0.1$ | $65.1 \pm 0.1$ | $51.5 \pm 0.1$ | $52.9 \pm 0.1$ | $53.3 \pm 0.1$ |
| MEDOIDS | $55.1 \pm 0.1$ | $65.0 \pm 0.1$ | $51.5 \pm 0.0$ | $52.9 \pm 0.1$ | $53.1 \pm 0.1$ |
| DPPNET MODE | $\mathbf{53.6 \pm 0.3}$ | $\mathbf{63.6 \pm 0.4}$ | $\mathbf{50.8 \pm 0.2}$ | $\mathbf{51.4 \pm 0.3}$ | $\mathbf{51.6 \pm 0.4}$ |

| DIGIT | 5 | 6 | 7 | 8 | 9 | All |
|---|---|---|---|---|---|---|
| DPP BASELINE | $50.4 \pm 0.1$ | $51.6 \pm 0.1$ | $51.5 \pm 0.1$ | $50.9 \pm 0.1$ | $52.7 \pm 0.1$ | $49.2 \pm 0.1$ |
| UNIF | $52.4 \pm 0.1$ | $54.6 \pm 0.1$ | $55.1 \pm 0.1$ | $53.3 \pm 0.1$ | $56.2 \pm 0.1$ | $51.6 \pm 0.1$ |
| MEDOIDS | $52.4 \pm 0.0$ | $54.4 \pm 0.1$ | $55.1 \pm 0.1$ | $53.2 \pm 0.1$ | $56.1 \pm 0.1$ | $51.0 \pm 0.1$ |
| DPPNET MODE | $\mathbf{51.8 \pm 0.3}$ | $\mathbf{52.8 \pm 0.3}$ | $\mathbf{52.7 \pm 0.4}$ | $\mathbf{50.9 \pm 0.3}$ | $\mathbf{55.0 \pm 0.4}$ | $\mathbf{48.6 \pm 0.2}$ |

Table 3: NLLs on CelebA and MovieLens (mean $\pm$ standard error); 20 samples of size 20 were drawn for 20 different feature matrices each, with 100 samples per method; DPPNET achieves the best NLLs.

| DATASET | KERNEL | DPP BASELINE | UNIFORM | $k$-MEDOIDS | DPPNET Mode |
|---|---|---|---|---|---|
| CelebA | RBF | $49.04 \pm 2.03$ | $50.84 \pm 1.53$ | $51.18 \pm 1.34$ | $\mathbf{49.28 \pm 1.57}$ |
| MovieLens | Linear | $84.29 \pm 0.20$ | $92.04 \pm 0.17$ | $88.90 \pm 0.16$ | $\mathbf{80.21 \pm 0.33}$ |

### 4.3 DPPNET for kernel reconstruction

As a final experiment, we evaluate DPPNET's performance on a downstream task for which DPPs have been shown to be useful: kernel reconstruction using the Nyström method [42, 48]. Given a positive semidefinite matrix $\mathbf{K} \in \mathbb{R}^{N \times N}$, the Nyström method approximates $\mathbf{K}$ by $\hat{\mathbf{K}} = \mathbf{K}_{\cdot,S} \mathbf{K}_{S,S}^{\dagger} \mathbf{K}_{S,\cdot}$ where $\mathbf{K}^{\dagger}$ denotes the pseudoinverse of $\mathbf{K}$ and $\mathbf{K}_{\cdot,S}$ (resp. $\mathbf{K}_{S,\cdot}$) is the submatrix of $\mathbf{K}$ formed by its rows (resp. columns) indexed by $S$. The Nyström method is a popular choice to scale up kernel methods, *e.g.*, [3, 45, 14, 46]. Reconstruction quality depends directly on the selected set of columns $S$; choosing the columns by sampling from the DPP with kernel $K$ is a standard approach [31, 38].

Following the approach of Li et al. [31], we evaluate the quality of the kernel reconstruction via the following process: given a RBF ridge regression kernel $\mathbf{K}$ built from 1000 training points in the Ailerons regression dataset, and with regularization and bandwidth chosen using 10-fold cross validation, we report the test prediction error obtained by the Nyström reconstruction $\hat{\mathbf{K}}$. The columns for the reconstruction are chosen with different DPP sampling algorithms: full DPP sampling, DPPNET and approximate DPP sampling using MCMC with quadrature acceleration [31, 32].

Fig. 5 reports our results, and confirms that DPPNET-based mode sampling performs comparably to other DPP sampling methods (Fig. 5a), while running orders of magnitude faster. Furthermore, while all methods were run on CPU, DPPNET is amenable to further acceleration using GPUs.

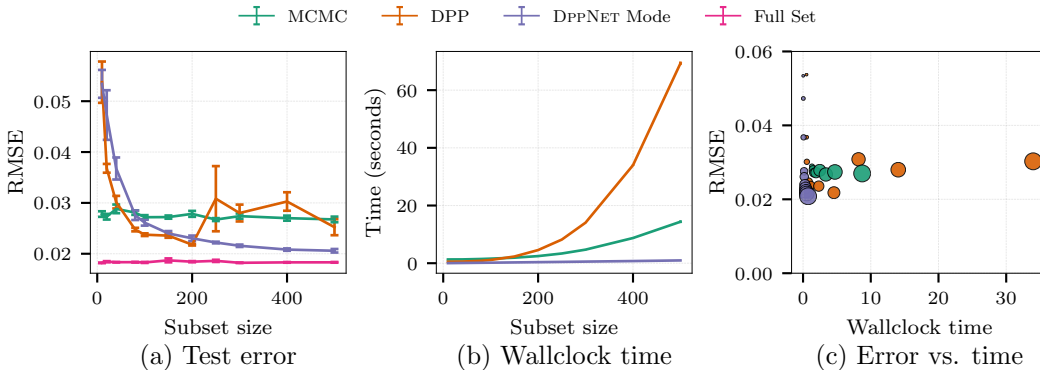

(a) Test error     (b) Wallclock time     (c) Error vs. time

Figure 5: Results for the Nyström approximation experiments, comparing DPPNET to the fast MCMC sampling method [32]. Subset selection by DPPNET achieves comparable or lower RMSE than other methods and is significantly faster. In (c), the relative size of the marker represents the size of the sampled subset.

## 5 Conclusion and discussion

We introduced DPPNET: a generative network that approximates DPP sampling over fixed and varying ground sets. We showed across several datasets and applications that DPPNETs are orders of magnitude faster than standard DPP sampling algorithms, without decreasing sample quality.

We derived an inhibitive attention mechanism based on the repulsion process modeled by DPPs; added to DPPNET while learning a class of DPPs over ground sets that vary over time, this mechanism significantly reduces the number of trainable parameters required to learn a DPPNET.

Using DPPNET, several applications of DPPs that remained purely theoretical in practice due to high sampling costs (*e.g.*, minibatch sampling for SGD as suggested in [51]) are now within reach of modern computing abilities; as such, replacing DPPs with DPPNET in cases where approximate, fast sampling is required in downstream applications is a key area for future work.

Our choice of architecture for DPPNET leaves certain questions open. DPPNETs samples are *not* exchangeable: two sequences $i_1, \ldots, i_k$ and $\sigma(i_1), \ldots, \sigma(i_k)$ where $\sigma$ is a permutation of $[k]$ will not have the same probability under a DPPNET. Although exchangeability can be enforced by leveraging previous work [50], non-exchangeability can be an asset when sampling a ranking of items.

Finally, our theoretical results (Thm. 1) suggests a new area of research in terms of using generative networks for combinatorial optimization. Two questions of particular interest are the following. Which properties of set functions, other than submodularity, can be inherited by a generative model? Can generative neural networks be leveraged to learn other combinatorial functions for which marginal probabilities (used to train the network) can be easily obtained?

**Acknowledgements.** The authors would like to thank D. Sculley and Dustin Tran for their help with this project.

## Footnotes

[3]The expected sampled set size under a DPP depends on the eigenspectrum of the kernel $\mathbf{L}$.

[4]Such tasks require the model to output a set of likely items given a pre-selected choice of items, for example when recommending items to customers that have already chosen certain items to purchase.

[5]This inner product could be replaced by the kernel function that defines the true DPP for DPPNET.

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

## A Maintaining log-submodularity in the generative model

**Theorem 1.** *Let $f$ be a strictly submodular function over subsets of a ground set $\mathcal{Y}$, and $g$ be a function over the same space such that*

$$\|f - g\|_\infty \leq \min_{\substack{S \neq T \\ S,T \notin \{\emptyset, \mathcal{Y}\}}} \frac{1}{4} \left[ f(S) + f(T) - f(S \cup T) - f(S \cap T)) \right]. \tag{4}$$

*Then $g$ is also submodular.*

*Proof.* In all the following, we assume that $S, T$ are subsets of a ground set $\mathcal{Y}$ such that $S \neq T$ and $S, T \notin \{\emptyset, \mathcal{Y}\}$ (the inequalities being immediate in these corner cases). Let

$$\epsilon := \min_{S,T} f(S) + f(T) - f(S \cup T) - f(S \cap T))$$

By the strict submodularity hypothesis, we know $\epsilon > 0$.

Let $S, T \subseteq \mathcal{Y}$ such that $S \neq T$ and $S, T \neq \emptyset, \mathcal{Y}$. To show the log-submodularity of $g$, it suffices to show that

$$g(S) + g(T) \geq g(S \cup T) + g(S \cap T).$$

By definition of $\epsilon$,

$$f(S) + f(T) - f(S \cup T) - f(S \cap T)) \geq \epsilon$$

From equation 4, we know that

$$\max_{S \subseteq \mathcal{Y}} |f(S) - g(S)| \leq \epsilon/4.$$

It follows that

$$g(S) + g(T) - g(S \cup T) + g(S \cap T)$$
$$\geq f(S) + f(T) - f(S \cup T) - f(S \cap T) - \epsilon$$
$$\geq 0$$

which proves the submodularity of $g$. $\qquad\square$

## B Encoder details

For the MNIST encodings, the VAE encoder consists of a 2d-convolutional layer with 64 filters of height and width 4 and strides of 2, followed by a 2d convolution layer with 128 filters (same height, width and strides), then by a dense layer of 1024 neurons. The encodings are of length 32.

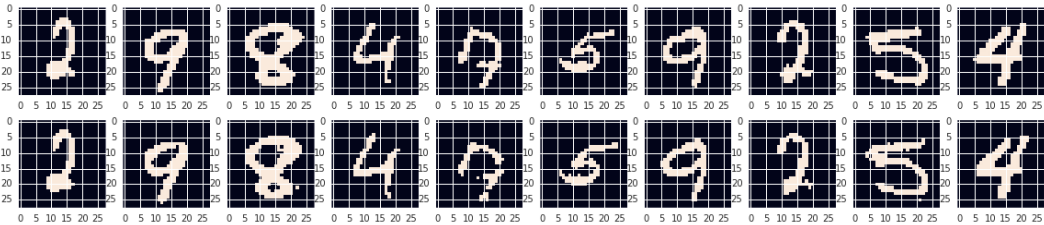

Figure 6: Digits and VAE reconstructions from the MNIST training set

CelebA encodings were generated by a VAE using a Wide Residual Network [49] encoder with 10 layers and filter-multiplier $k = 4$, a latent space of 32 full-covariance Gaussians, and a deconvolutional decoder trained end-to-end using an ELBO loss. In detail, the decoder architecture consists of a 16K dense layer followed by a sequence of $4 \times 4$ convolutions with $[512, 256, 128, 64]$ filters interleaved with $2\times$ upsampling layers and a final $6 \times 6$ convolution with 3 output channels for each of 5 components in a mixture of quantized logistic distributions representing the decoded image.

