[Reviews · NeurIPS 2019]

Reviewer 1



This work focuses on sampling determinantal point processes (DPPs) by utilizing deep learning architecture so that the running time for sampling becomes faster than the standard DPP sampling. To avoid complex operations (e.g., matrix inversion or eigendecomposition) used in the standard DPP sampling, authors come up with an attention mechanism where it simply needs to dot-product of features or kernel matrix itself. As far as I know, most of the prior works in deep learning community borrow DPPs and apply them at the top of layer in the network for the purpose of generalization, compression and so on. Unlike the priors, this work utilizes deep learning models in order to simplify the DPP sampling. This can be a great impact for future works of other models whose inputs are a subset of items and guide them to capture their characteristic (e.g., ranking) using deep learning architectures. Authors also analyze that the proposed network satisfies with the log-submodularity under certain condition, which can justify using the greedy algorithm for approximating the mode. Although the condition is even hard to check in practical, empirical results for the mode take the best performance in practical applications including kernel reconstruction. This paper is also well-written and well-organized. The motivations and contributions are intuitive and straightforward. But, some descriptions may give confusion to readers not familiar with DPPs. For example, the objective function to train the model is not provided formally. Authors should describe the objective function (or loss) for DPP sampling more concretely. And it is ambiguous how the normalized log-likelihood and negative log-likelihood are different. As they reported in the experiment, samples of DPP mode from their network can achieve the minimum NLLs. However, it is not fair to compare their DPP mode to other sampling-based methods, but not for mode. It would be better to benchmark the standard DPP mode algorithm (a.k.a.DPP MAP inference). In addition, even the sampling process is simple and fast, the training time to learn the model is essential. There are also works for boosting DPP sampling algorithm, e.g., tree-based algorithm [3] or Nymstrom approximation [4]. Both can take the sublinear-time with respect to the size of the ground set while the proposals are linear-time, which can be slower for the large-scale setting. It would be also better to compare the running time of the proposal to other fast algorithms. Although their benchmarks are weak due to a lack of other similar works, their approach is a new concept applying for DPP and valuable for other works related to DPP. Therefore, it is enough to accept. [3] Jennifer Gillenwater, Alex Kulesza, Zelda Mariet, Sergei Vassilvitskii. “A Tree-Based Method for Fast Repeated Sampling of Determinantal Point Processes”, ICML 2019. [4] Michał Dereziński, Daniele Calandriello: “Exact sampling of determinantal point processes with sublinear time preprocessing”, 2019, arXiv:1905.13476. ========================================================================================== I read all reviews and author feedback. Authors have replied my comments with clear and promising explanations. The contributions of this paper would be further improved if all comments come up with the final manuscript.

Reviewer 2



** Proviso ** I wanted to pre-emptively point out that besides a general familiarity with the standard formulation and use of DPPs, I do not consider myself up-to-date with recent developments in this field. Consequently, I may not give the most critical assessment of the work. However, I will do my best to review the work to the best of my knowledge. -- Paper Summary -- DPPs are a well-established family of models for obtaining a set of diverse samples from a fixed or varying ground set. However, the widespread use of such models is hampered by their computational complexity, which is tied to the inversion or decomposition of a possibly large kernel matrix. In this work, the authors propose an alternative DPP model which is instead formulated as a neural network. The proposed architecture is shown to satisfy properties that are typically expected of DPPs, while also providing a substantial speed-up in comparison to standard DPP models and more recent alternatives based on MCMC sampling. The effectiveness of this method is showcased using a synthetic experiment for drawing samples from a unit square, a comparison to other techniques on more widely-established datasets, as well as a practical application of DPPs for kernel reconstruction. The scope of the paper covers both theoretical contributions, as well as more practical elements for designing and fine-tuning the architecture of the proposed DPPNet. -- Writing/Clarity -- The paper is extremely well written and structured. It was a pleasure to read, and I could barely find any typos in the text. For someone who does not follow recent developments on the topic of DPPs, I found that the concepts explained in the paper are well conveyed and easy to follow. The figures are also neat and properly complemented by the associated text. Good job! Some minor typos: - L71: ‘k’ is defined in the abstract, but not in the main text. While fairly obvious, reintroducing it would be better; - L104: Lower case ’n’ hadn’t been used prior to this instance; - L131: Extra ‘a’; L133: ‘with’ -> ‘where’; - L182: Corollary is shortened as ‘Cor.’ elsewhere. -- Originality and Significance -- Using a neural network for emulating the functionality of a DPP appears to be a novel contribution which has not previously appeared in the literature. Beyond simply explaining a basic model for carrying out this task, the paper also presents several variations including an attention mechanism, an algorithm for greedy sampling, and also points out several workarounds for overcoming potential limitations associated with this method (such as setting a buffer N’ > N in case of having a ground set which can increase in size). To the best of my understanding, these are all noteworthy contributions which go beyond being simply ‘incremental’. I think there is an appealing depth to this work which balances both theoretical contributions - in terms of showing how DPPNets obey certain properties of DPPs - as well as more practical ‘engineering’ elements which enable the speed-ups highlighted in the experiments. Perhaps the paper slightly struggles in coming up with a convincing real-world problem where the increased scalability of DPPNet is essential, but the speed-ups achieved in relation to other methods without compromising on performance should definitely interest practitioners woking with DPPs. The theoretical aspects of the paper, although interesting, are not explored thoroughly, which the authors put down to the intractability of effectively verifying the provided proofs. Nonetheless, I think this sets up an interesting avenue for future work by encouraging further exploration of this connection between neural networks and DPPs. -- Technical Quality/Evaluation -- As highlighted in the preface to this review, I cannot vouch for the correctness of certain theoretic aspects of the work presented here. However, the material I was able to verify appeared to be correct and clearly presented. The experimental evaluation is also well rounded, highlighting the different variations of DPPNet in comparison to competing techniques before ultimately settling upon a single variation (DPPNet Mode) which performs more consistently than the standard DPPNet. The experiments seem fair, with appropriate metrics (including variance) provided for each set-up. The results should also be reproducible since there is sufficient material in the section (and followed up in the appendix) which includes details on how the datasets were preprocessed and how the architectures were constructed. -- Overall recommendveration -- While re-iterating the proviso to undertaking this review, I do genuinely believe that this to be a very well-rounded paper which ticks most of the criteria I would expect from a high quality submission. The paper is written in an exemplary manner, the concepts are clearly explained, and the experiments supplement the novelty of the proposed method by also showing how its use can lead to both speed-ups and also performance improvements over competing DPP methods. ** Post-rebuttal update ** Thank you for your reply. I liked this paper from the first read-through, and reading the other reviews and your rebuttal confirmed my original impressions. There are some minor refinements suggested by the other reviewers which could further improve the overall quality of the submission, such as updating the related work section with references to the recently published work indicated in another review, clarifying parts of the exposition, including the additional comparison etc. However, this remains a solid paper through and through._x000C_

Reviewer 3



Originality: The idea for applying neural network with an attention mechanism to approximate DPP probability seems to be new but the novelty of the idea seems to be limited as it just applies existing techniques to a new scenario without a novel adaptation to it. The theoretical justification in the main paper is more novel. Quality: The technique used in this paper seems to be valid and the objective function is validated theoretically in a sense. The 3 experiments can show that the DPP net gives a reasonable approximations to true DPP. Clarity: This paper is clearly written and relatively easy to follow. However, it will be very helpful to provide a slightly more detailed review of DPP in the appendix for non-expert readers. Significance: The key contribution of DPP net is to reduce the computational time for DPP to linear (excluding the training time). This seems to be valuable for practitioners that allows them to use DPP for real, large data set. This can also act as a component for downstream tasks. My main criticism is the novelty.

[Author Response · NeurIPS 2019]

Thank you for your detailed reviews and comments. We hope our clarifications, which we will include in the final version of the paper, will strengthen your confidence in the novelty and significance of our results. We begin by addressing two crucial concerns that were raised.

*Novelty.* We believe that the following results are significantly novel:

(a) Theory: we prove generative models can recover the sub- and supermodularity of target distributions. These are fundamental properties for combinatorial optimization, and as such this result is an important step for the theoretical analysis of generative models over discrete spaces.

(b) Algorithms: DPPNET sampling is the only DPP sampling algorithm to generalize to new data without requiring updates to pre-computed information. Even very recent work [2, 3] (published after NeurIPS) requires pre-processing that relies on the immutability of the DPP kernel. In comparison, DPPNET can draw samples from new kernels as long as the feature representations of the new items are drawn from the same distribution as the training data. This significantly increases the scope of application for DPPs.

(c) Experiments: we show that current neural architectures are, under the right training conditions, able to represent DPPs to a degree of precision sufficient to replace DPPs in downstream applications.

*Comparison to MAP.* We will update our work to include NLL results for the MAP approximations for standard DPPs; we do not expect DPPNET to outperform the DPP mode. Although DPPNET mode has the same complexity as DPPNET sampling, the same does not hold for standard sampling (in particular, [4] and [1] grow as $\mathcal{O}(N^3)$ and hence will be slower than MCMC sampling [5] for which we provide a timing comparison). Since standard DPP sampling costs $\mathcal{O}(N^3)$, our timing results for standard sampling on the Nystrom experiments provide a lower bound on how much acceleration we can expect over previous MAP inference algorithms.

**Reviewer 4.** Thank you for your review and your comment about MAP algorithms. We hope our above clarification answers your question; we will update our paper to clarify this important point.

*Objective function.* We will write out the objective function explicitly and clarify the NLL notation.

*Fast sampling related work ([2, 3]).* These works (made available online after the NeurIPS submission deadline) are indeed highly relevant. Both speed up DPP sampling given a polynomial time pre-processing step. However, this pre-processing needs to be re-applied when the ground set is changed unless the change belongs to a specific family of transformations [3]. This is not the case of DPPNET. For this reason, [2, 3] are complementary to DPPNET; DPPNET will be more efficient when the true DPP changes overtime, but [2, 3] should be preferred for fixed kernels. We will gladly update our work to include this discussion.

To be more precise, the Nystrom approximation of [2] has to be computed every time the ground set changes. If the ground set changes frequently, this is prohibitively expensive as soon as $k \geq 5$, costing $\mathcal{O}(Nk^6 \log^2 \frac{N}{\delta} + k^9 \log^3 \frac{N}{\delta})$ [2, Thm 1 for DPPs, page 9]. The tree construction [3] can be pre-processed only if samples are drawn from DPPs whose kernels are of the form $L = B^\top WWB$ with fixed $B$ and varying diagonal $W$. In comparison, as long as the new features are drawn from the same distribution as the training data, we show experimentally that DPPNET generates high quality samples without requiring re-training or additional pre-processing.

**Reviewer 5.** Thank you for the kind review — as recommended, we focus on concerns raised by other reviewers.

*Applications.* Applications of DPPs to problems in ML have been limited by the poly($N$) cost of sampling when the ground set varies often (*e.g.*, certain recommender systems settings). DPPNET is a viable method to address such obstacles, and applying DPPNET to such problems is planned future work.

**Reviewer 6.** Thank you for the detailed comments; we have summarized the key novel contributions of our work at the beginning of the rebuttal; we will be certain to emphasize these in the final version of our paper.

*Line 151.* We will clarify this. We mean that if the kernel for training data has an expensive computational cost (*e.g.*, needs to be learned from data), this expensive computational cost will only be required during training, and not when generalizing to new or updated datasets.

*Training objective/algorithm.* We will clarify this.    *DPP literature review in appendix.* We will add this.

[1] L. Chen, G. Zhang, and E. Zhou. Fast greedy map inference for DPP to improve recommendation diversity. In *NeurIPS*. 2018.

[2] M. Dereziński, D. Calandriello, and M. Valko. Exact sampling of DPPs with sublinear time preprocessing, 2019.

[3] J. Gillenwater, A. Kulesza, Z. Mariet, and S. Vassilvtiskii. A tree-based method for fast repeated sampling of DPPs. In *ICML*, 2019.

[4] I. Han, P. Kambadur, K. Park, and J. Shin. Faster greedy MAP inference for determinantal point processes. In *ICML*, 2017.

[5] C. Li, S. Jegelka, and S. Sra. Fast DPP sampling for Nystrom with application to kernel methods. In *ICML*, 2016.


[Meta-Review · NeurIPS 2019]

All reviewers appreciate the quality of this contribution, and the clarifications in the rebuttal, particularly with regards to the differences between DPPNET and DPP MAP inference. Please pay careful attention to the reviewer comments in preparing your final revisions -- there was the sense the paper could be significantly improved by accounting for the discussions in the rebuttal and the questions and comments in the reviews.